# Cysteine and Folate Metabolism Are Targetable Vulnerabilities of Metastatic Colorectal Cancer

**DOI:** 10.3390/cancers13030425

**Published:** 2021-01-23

**Authors:** Josep Tarragó-Celada, Carles Foguet, Míriam Tarrado-Castellarnau, Silvia Marin, Xavier Hernández-Alias, Jordi Perarnau, Fionnuala Morrish, David Hockenbery, Roger R. Gomis, Eytan Ruppin, Mariia Yuneva, Pedro de Atauri, Marta Cascante

**Affiliations:** 1Department of Biochemistry and Molecular Biomedicine & Institute of Biomedicine of Universitat de Barcelona, Faculty of Biology, Universitat de Barcelona, 08028 Barcelona, Spain; jtarragocelada@ub.edu (J.T.-C.); cfoguet@ub.edu (C.F.); mtarrado@ub.edu (M.T.-C.); silviamarin@ub.edu (S.M.); xa.he.al@gmail.com (X.H.-A.); jperarnau33@gmail.com (J.P.); pde_atauri@ub.edu (P.d.A.); 2Centro de Investigación Biomédica en Red de Enfermedades Hepáticas y Digestivas (CIBEREHD) and Metabolomics Node at Spanish National Bioinformatics Institute (INB-ISCIII-ES-ELIXIR), Instituto de Salud Carlos III (ISCIII), 28020 Madrid, Spain; 3Fred Hutchinson Cancer Research Center, Seattle, WA 98109, USA; fmorrish@fredhutch.org (F.M.); dhockenb@fredhutch.org (D.H.); 4ICREA, 08010 Barcelona, Spain; roger.gomis@irbbarcelona.org; 5Institute for Research in Biomedicine Barcelona (IRB Barcelona) and The Barcelona Institute of Science and Technology, 08028 Barcelona, Spain; 6CIBERONC, Instituto de Salud Carlos III (ISCIII), 28020 Madrid, Spain; 7Department of Medicine, Faculty of Medicine, Universitat de Barcelona, 08036 Barcelona, Spain; 8Center for Cancer Research, Cancer Data Science Laboratory, National Cancer Institute, Bethesda, MD 20892, USA; eytan.ruppin@nih.gov; 9Oncogenes and Tumour Metabolism Laboratory, The Francis Crick Institute, London NW1 1AT, UK; mariia.yuneva@crick.ac.uk

**Keywords:** colorectal cancer, metastasis, redox metabolism, genome-scale metabolic models

## Abstract

**Simple Summary:**

In this work, we studied the metabolic reprogramming of same-patient-derived cell lines with increasing metastatic potential to develop new therapeutic approaches against metastatic colorectal cancer. Using a novel systems biology approach to integrate multiple layers of omics data, we predicted and validated that cystine uptake and folate metabolism, two key pathways related to redox metabolism, are potential targets against metastatic colorectal cancer. Our findings indicate that metastatic cell lines are selectively dependent on redox homeostasis, paving the way for new targeted therapies.

**Abstract:**

With most cancer-related deaths resulting from metastasis, the development of new therapeutic approaches against metastatic colorectal cancer (mCRC) is essential to increasing patient survival. The metabolic adaptations that support mCRC remain undefined and their elucidation is crucial to identify potential therapeutic targets. Here, we employed a strategy for the rational identification of targetable metabolic vulnerabilities. This strategy involved first a thorough metabolic characterisation of same-patient-derived cell lines from primary colon adenocarcinoma (SW480), its lymph node metastasis (SW620) and a liver metastatic derivative (SW620-LiM2), and second, using a novel multi-omics integration workflow, identification of metabolic vulnerabilities specific to the metastatic cell lines. We discovered that the metastatic cell lines are selectively vulnerable to the inhibition of cystine import and folate metabolism, two key pathways in redox homeostasis. Specifically, we identified the system xCT and MTHFD1 genes as potential therapeutic targets, both individually and combined, for combating mCRC.

## 1. Introduction

Colorectal cancer is the second leading cause of cancer mortality with over 800,000 deaths per year [1]. Surgery is usually the primary treatment and is successful in approximately 50% of patients [2]. Disease recurrence after surgery involving distant metastasis, frequently in the liver, is a major problem and is often the ultimate cause of death [3]. At that stage, only 25% of colorectal cancer patients with isolated liver metastasis benefit from a multimodal treatment including surgery. Indeed, with current therapies, the 5-year-survival rate of patients with metastatic colorectal cancer (mCRC) is less than 10% [4]. Thus, it is of paramount importance to develop effective therapeutic strategies against colorectal cancer metastasis.

Recently, metabolic reprogramming has emerged as an enabling feature of metastasis and cancer stemness [5]. Briefly, metastatic spread requires the ability to cope with the increased formation of reactive oxygen species (ROS) and the metabolic plasticity to adapt to a variable supply of substrates in the secondary tumour site [6,7]. Indeed, there is emerging evidence that exploring and targeting the specific metabolic rewiring that supports metastatic activity could be exploited for therapy [8,9,10].

Here we have used a unique set of same-patient-derived cell lines with increasing metastatic potential (SW480, SW620 and SW620-LiM2) to identify druggable metabolic vulnerabilities against mCRC. SW480 and SW620 are KRAS^G12V^-mutant cell lines derived from the primary tumour and a lymph node metastasis, respectively, of a patient with a Duke’s type B colorectal cancer [11], whereas SW620-LiM2 is a liver metastatic-enriched cell line derivative from SW620 [12], hereafter referred to as LiM2. Using a newly developed workflow, we integrated experimental data (i.e., extracellular flux measurements, metabolomics, ^13^C stable resolved metabolomics, transcriptomics, respiration parameters, growth rate, and gene dependencies) to build cell line-specific Genome Scale Metabolic Models (GSMMs). 

This developed workflow enable us to study the metabolic phenotype that emerges from the complex multilevel metabolic and genetic interactions and to identify metabolic vulnerabilities that might not be apparent from a single type of data [13,14]. In this regard, the analysis of cell line specific GSMMs pointed to cysteine and folate metabolism as key players in the metabolic rewiring of the metastatic cell lines and identified the cysteine/glutamate transporter (system xCT) and methylenetetrahydrofolate dehydrogenase 1 (MTHFD1) as putative targets. We validated the predicted targets in vitro using specific inhibitors for system xCT and MTHFD1. These inhibitors had a more significant effect on metastasis-derived cell lines versus the primary tumour-derived cell line and only a minor effect on non-tumour NCM460 cells. NCM460 is a cell line derived from colon mucosal epithelium [15], which has been previously validated as a good cell model to be used as a control in the development of new therapeutic strategies against colorectal cancer [16]. Overall, our results highlight the importance of cysteine and folate metabolism for metastatic cell growth and the therapeutic potential of inhibiting both pathways to target mCRC.

## 2. Results

### 2.1. Characterisation of the Metastatic Phenotype

To gain better understanding of the metabolic phenotype associated with metastatic potential in colon cancer, we used cell lines differing in malignancy potential and characterised their growth, expression of metabolic modulators, invasion and tumorigenic capacities. The primary tumour-derived SW480 was the slowest-growing cell line, followed by the metastatic LiM2 and SW620 cell lines (Figure 1a and Appendix A). Increased proliferation in metastatic cell lines was correlated with increased MYC protein levels, while P-AKT levels were similar in the three cell lines (Figure 1b, Appendix A). Cell volume was inversely correlated with cell proliferation (Figure 1c), and this was coupled to differences in mTOR signalling and protein content (Figure 1b, Appendix A). The metastatic potential of SW620 and LiM2 had been previously established in vivo by Urosevic et al. [12], and both cell lines were capable of forming liver and lung metastases following splenic injection in mice, but LiM2 was determined to be significantly more metastatic. As expected, the metastatic cell lines also showed higher 3D growth capacity in vitro than the primary cell line SW480, demonstrated by the increased spheroid area (Figure 1d), and LiM2 exhibited the highest capacity to form secondary spheroids (Figure 1e). 

Remarkably, wound healing assays indicated that the primary cell line SW480 had a higher migratory capacity than the metastatic cell lines (Figure 1f and Appendix A), consistent with the fact that proliferation and migration gene expression programs have been reported to be different [17]. Examining markers of epithelial mesenchymal transition (EMT), we found that SW480 displayed reduced E-cadherin (Figure 1g and Appendix A) and increased EMT transcription factor ZEB1/2 (Figure 1g and Appendix A). However, the levels of other mesenchymal markers (e.g., N-cadherin and vimentin) and transcription factors associated with EMT (e.g., Twist 1/2, SNAI1 and NF-κB) were increased in SW620 and LiM2 cells, with remarkably high protein levels of SNAI1 in the latter.

### 2.2. The metastatic Cell Lines Display Increased Glucose, Glutamine and Mitochondrial Metabolism

Analysing metabolic differences between the cell lines, we demonstrated that the metastatic cell lines (SW620 and LiM2) had a stronger Warburg effect than SW480, as they consumed more glucose and produced more lactate (Figure 2a,b). Subsequent analysis of glucose entrance into the tricarboxylic acid (TCA) cycle, evaluated by [1,2-^13^C]-glucose incubation, indicated a higher glucose contribution into the TCA in the metastatic cell lines (Figure 2c,j). Consistent with this result the levels of phosphorylated pyruvate dehydrogenase (PDH) were decreased in metastatic cells (Figure 2d and Appendix A) indicating its higher activity.

The second major substrate used by cancer cells is glutamine. The metastatic cells consumed more glutamine and produced more glutamate (Figure 2e,f). Furthermore, we observed an increased incorporation of [U-^13^C]-glutamine into the TCA cycle through both oxidative and reductive pathways in the metastatic cell lines (Figure 2h,i,k). Increased uptake of glutamine in these cell lines correlated with the increase in protein levels of key glutamine catabolising enzymes glutaminase (GLS) and glutamate dehydrogenase 1 (GLUD1) (Figure 2g and Appendix A).

In line with the metabolic changes observed above, mitochondrial function was also enhanced in both SW620 and LiM2, which displayed an increased oxygen consumption rate (OCR) compared to SW480 (Appendix A). The titrations with glucose, glutamine and palmitate corroborated an increased OCR in the metastatic cell lines (Appendix A). Mito Fuel assays, performed with specific inhibitors in order to block the utilisation of the three major respiratory substrates, revealed that the metastatic cell lines had similar capacity but lower dependency and higher flexibility for the three substrates, when compared to the SW480 cell line (Appendix A).

To complete the characterisation of metabolic traits associated with metastatic capacity, we measured intracellular metabolite concentrations as well as metabolite exchange fluxes between cells and cell culture media using targeted metabolomics and HPLC/MS/MS (Appendix A). Intracellular glutamate and glutamine concentrations were significantly higher in metastatic cell lines (SW620 and LiM2) than in SW480, consistent with their higher rate of glutamine uptake and glutaminolysis (Figure 2e,i). Moreover, we observed increased intracellular glycine concentration (Appendix A), and increased serine consumption and reduced glycine secretion in the metastatic cell lines (Appendix A). The latter results suggested increased serine hydroxymethyltransferase (SHMT) activity and thus enhanced folate metabolism. Interestingly, metastatic cells displayed decreased intracellular concentrations and increased consumption of the essential amino acid (EAA) pools, an indicator of either enhanced protein synthesis or increased catabolism of EAA to fuel the TCA and respiratory chain (Appendix A).

Additionally, acyl-carnitines and free carnitines were detected at significantly higher concentrations in SW480 compared to metastatic cell lines (Appendix A), potentially indicative of differences in the metabolism of branched-chain amino acids and lipids. Finally, the metastatic cell lines produced more polyamines, which is consistent with an increased arginine consumption (Appendix A).

### 2.3. The Metabolic Adaptation of the Metastatic Cell Lines Observed In Vitro Is Maintained in an In Vivo Scenario

A significant concern when working with in vitro models is that their metabolic phenotype can be significantly different from those encountered in vivo [18]. To determine if key features of our cellular models were conserved in vivo we compared metabolism of the three cell lines grown as xenografts in NOD/SCID mice. We observed that tumour growth rate in mice correlated with the proliferation rate and metastatic phenotype observed in vitro (Figure 1a,d and Figure 3a). No metastases were detected in any case (Appendix A), likely because mice had to be culled before they could develop them. Moreover, an excellent agreement with in vitro cell data was also observed for EMT markers (e.g., E-cadherin, vimentin), GLS and PDH phosphorylation status (Figure 3b,c, Appendix A).

Although the contribution of both glucose and glutamine into the TCA cycle evaluated after a bolus injection of either [U-^13^C]-glucose or [U-^13^C]-glutamine was higher in vivo in comparison with in vitro (Figure 3d–f), the isotopologue labelling patterns were comparable for both conditions. All together, these data suggest a similar metabolic behaviour in in vitro and in vivo models. This indicates that any metabolic targets identified in the in vitro cell models would be independent of the tumour microenvironment to which tumour cells are exposed and would likely be recapitulated in vivo.

### 2.4. Computational Inference of Cell Line-Specific Metabolic Flux Maps and Metabolic Targets through Multiomics Data Integration

In order to obtain predictive cell line-specific flux maps for SW480, SW620, and LiM2, we integrated the measured growth rates, rates of metabolite uptake and secretion, respiration parameters, ^13^C resolved metabolomics, and targeted metabolomics using a newly developed workflow (Figure 4a). Transcriptomics [12,19] and gene dependencies of the subset of metabolic genes analysed in the Project DRIVE (deep RNAi interrogation of viability effects in cancer) database [20] were also integrated.

As expected, the resulting cell line-specific flux maps showed significantly higher glycolytic flux in the metastatic cell lines compared to SW480, as exemplified by the flux through hexokinase 1 and lactate dehydrogenase (Figure 4b). Furthermore, consistent with the phosphorylation status of PDH, the flux maps presented a higher flux through PDH and citrate synthase in SW620 and LiM2 than in SW480. It is worth noting that the computed flux maps showed that in the metastatic cell lines, the PDH/lactate dehydrogenase flux ratio was still quite low (roughly 15%), suggesting that despite higher PDH activity, the metastatic cell lines still predominantly relied on aerobic glycolysis. Likewise, cell line-specific flux maps predicted increased glutaminase activity in the metastatic cell lines, and in LiM2 in particular, in concordance with the increased glutaminase levels and glutamine consumption rates (Figure 4b). To elucidate whether these differences were intrinsically linked to the enhanced proliferation of the metastatic cell lines, fluxes were also expressed relative to the growth rate (Appendix A). Remarkably, while TCA-associated fluxes (i.e., PDH, citrate synthase and glutaminase) appeared to be closely associated with the proliferation rate, the relative glycolytic flux was still significantly higher in the metastatic lines, suggesting that the latter may play a role in the metastatic phenotype that goes beyond supporting cellular proliferation. 

The cell line-specific flux maps were used to systematically simulate genes’ KO using the Minimization of metabolic adjustment (MOMA) algorithm [21] and identify single or target pairs that could selectively inhibit growth in the metastatic cell lines. Overall, 10 single target and 237 target combinations were predicted to impair the proliferation of SW620 and LiM2 (Appendix A). Of these targets, we focused on targets related to either folate or cysteine metabolism as they were predicted to be the most selective against metastatic cells (Table 1).

### 2.5. Metastatic Cell Lines Are Dependent on Cysteine Uptake and Vulnerable to System xCT and Glutathione Reductase Inhibition

The simultaneous inhibition of the two main cystine transport systems, the cystine/glutamate transporter system xCT (coded by the genes SLC7A11 and SLC3A2) [22] and the cystine/neutral amino acid antiport acid system b0,+ (coded by the genes SLC7A9 and SLC3A1) [23] was predicted as selective for the metastatic cell lines (Table 1). Likewise, glutathione reductase (GSR), which catalyses the reduction of oxidised glutathione, was also predicted as a selective target (Table 1). Cysteine is the limiting substrate of glutathione synthesis [24] and can be derived from methionine through the transsulfuration pathway [25]. Alternatively, cysteine can be produced from the reduction of cystine in the cytoplasm, primarily by reacting with glutathione [26] but also by the thioredoxin reductase system [27], making all these putative targets interdependent (Figure 5a). Our GSMM analysis revealed that the metastatic cell lines had both insufficient activity of the transsulfuration pathway and of thioredoxin-dependent cystine reduction. Thus, the metastatic cell lines were predicted to be largely cystine/cysteine auxotroph, and dependent on both cystine carriers and GSR activity (Figure 5a).

To validate the predicted dependence on cystine uptake, we first incubated SW480, SW620, and LiM2 without cystine. We observed that under cystine deprivation, proliferation was more significantly reduced in the metastatic cell lines, confirming that they were more dependent on cystine uptake from the media (Figure 5b). As expected, cell proliferation was rescued through the addition of N-acetyl cysteine (NAC) which can be deacylated to form cysteine [28]. Next, we evaluated the therapeutic potential of inhibiting cystine transporters and, because simulations showed significantly higher flux through the system xCT (Figure 5c), we chose to focus on targeting it. With this aim, we evaluated the effects of two system xCT inhibitors: sulfasalazine, a drug approved for the treatment of rheumatoid arthritis [29], and erastin, a recently developed inhibitor of the system xCT [30,31]. As expected, both drugs had lower IC_50_ values for the metastatic cells than for SW480. Moreover, erastin exhibited IC_50_ values up to three orders of magnitude lower than those of sulfasalazine (Figure 5d,e and Appendix A). In addition, erastin also induced significant apoptosis in the metastatic cell lines and decreased 3D growth capacity (Appendix A). To further confirm the selectivity of these compounds towards the metastatic cells, we also evaluated their effect on a non-tumour colon NCM460 cell line, which is a cell line derived from healthy mucosa that has no spheroid-formation capacity (Appendix A). NCM460 cells had much lower sensitivity towards both of the compounds than the metastatic cells (Figure 5f,g and Appendix A). 

Next, to evaluate GSR as putative target, we used 2-AAPA, an inhibitor of GSR that has shown anticancer activity in many cancer cell lines [32,33,34]. In our cell model, 2-AAPA had lower IC_50_ values for the metastatic cell lines for the range of concentrations described in the literature (Figure 5f and Appendix A) with mildly or non-significant effects on apoptosis and 3D growth (Appendix A). NAC was able to rescue proliferation of the cell lines treated with 20 μM of 2-AAPA (Appendix A) but not at higher doses. Combining GSR and cystine transport inhibition demonstrated synergetic antiproliferative effects for the metastatic cell lines when first incubating with erastin for 72 h, and then adding 2-AAPA for a total duration of 120 h (Appendix A).

### 2.6. The Metastatic Cell Lines Are Vulnerable to Inhibition of Folate Metabolism

Our model predicted that the SW620 and LiM2 cell lines displayed significantly higher fluxes through the cytosolic folate pathway and were thus vulnerable to the inhibition of the cytosolic enzyme MTHFD1 (Table 1), which catalyses several steps of the cytosolic folate pathway (Figure 6a,b). The model specifically identified that, in the metastatic cell lines, the inhibition of the cytosolic folate pathway could not be compensated by the generally redundant folate mitochondrial pathway, because the CHO-THF generated by the mitochondrial isoenzyme (MTHFD2) could not be transported to the cytosol to compensate for MTHFD1 deficiency (Figure 6a). Therefore, in the metastatic cell lines, folate metabolism was, to some extent, uncoupled between the cytosol and the mitochondrial matrix, which would render them vulnerable to cytosolic folate pathway inhibitors.

To confirm the dependency on the cytosolic folate pathway, the cell lines were incubated with LY345899, an inhibitor of both MTHFD1 and MTHFD2 that has a significantly lower Ki for the former [35]. LY345899 was previously tested on the SW480/SW620 model and was reported to be selective for SW620 cells [36]. Our validation confirmed the prior reports and showed that the inhibitor was selective, not only for SW620, but also for the highly metastatic derivative of SW620-LiM2, while having little effect on the proliferation of NCM460 healthy colon epithelia cells (Figure 6c). Remarkably, the inhibitor did not induce apoptosis, suggesting that it primarily acted on cell proliferation (Appendix A). Additionally, LY345899 also inhibited spheroid formation in the metastatic cell lines (Appendix A).

Next, we evaluated the anti-proliferative effect of the antifolate-agent methotrexate which targets the cytosolic activity of dihydrofolate reductase (DHFR) [37]. Methotrexate displayed greater growth inhibitory effects in the metastatic cell lines compared to the SW480 and NCM460 cell lines (Appendix A). Similar to our findings with inhibition of MTHFD1, DHFR inhibition affected cell proliferation, and only a small fraction of cells underwent apoptosis (Appendix A). Furthermore, spheroids formation was strongly impaired in the metastatic cell lines under methotrexate treatment, reinforcing the importance of folate metabolism for metastatic colonisation (Appendix A). Next, we targeted both the cytosolic and the mitochondrial folate pathways using SHIN2 [38], a chemical inhibitor of both SHMT1 and SHMT2. We observed a similar growth inhibitory effect on both the primary and metastatic tumour cell lines but a significantly lower growth inhibitory effect on NCM460 at higher SHIN2 concentrations (Figure 6d, Appendix A).

Finally, as folate metabolism plays a key role in nucleotide synthesis, we evaluated whether the enhanced activity of folate metabolism in the metastatic cell lines could be attributed to their increased proliferation rate. We determined that, similarly to the Warburg effect, the enhanced fluxes through folate metabolism, particularly through the cytosolic branch, could not be solely attributed to the increased proliferative capacity of the metastatic cell lines (Appendix A).

### 2.7. Synergistic Effect of the Simultaneous Inhibition of Cysteine Uptake and Folate Metabolism

Having explored the inhibition of two major vulnerabilities of the metastatic cell lines predicted by the computational model (i.e., cysteine and folate metabolism), we evaluated whether the inhibition of these pathways could be synergetic. Indeed, both pathways have been associated with antioxidant defence as cystine contributes to glutathione synthesis, and MTHFD1 contributes to ROS detoxification by regenerating NADPH [39].

The results showed that the combination of erastin and LY345899 synergistically affected the proliferation of the metastatic cell lines from concentrations of 0.5 μM of erastin and 20 μM of LY345899 (Figure 7a–e and Appendix A). In line with this, the 3D growth of the SW620 cell line was significantly more impaired by the drugs in combination than with individual treatments. However, no differences were observed on spheroid formation in LiM2 cells between the combination and erastin individually due to the pronounced effect of the erastin treatment (Figure 7f,g). Additionally, the metastatic cell lines presented higher fractions of late apoptosis and necrosis with the combined drug treatment, but lower early apoptosis in comparison with erastin alone (Figure 7h).

## 3. Discussion

The cumulative evidence generated to date suggests that both the Warburg effect and glutaminolysis are associated with metastatic potential in several cancer types [8,9,40,41]. Adding to these data our study provides evidence that, when compared to the parental colon cancer cells (SW480), both lymph node- (SW620) and liver-derived (SW620-LiM2) metastatic cell lines displayed a significantly more active Warburg effect and increased incorporation of glucose and glutamine into the TCA cycle. In order to form macroscopic metastasis from a small number of seeding cells, metastatic cell populations must be endowed with a metabolic phenotype capable of supplying the building blocks and ATP to support rapid cell proliferation and thus, the increase in Warburg effect and glutaminolysis might enhance metastatic potential simply by enabling rapid cellular proliferation [7]. However, we determined that the glycolytic flux was enhanced in the metastatic cell lines far beyond their proliferation rate. This indicates that, in addition to cell proliferation, the Warburg effect could play a key role in other facets of the metastatic phenotype, such as endowing metastatic cells with the capacity to adapt to perturbations in the supply of both oxygen and glucose [42]. In this regard, the results generated from mitochondrial fuel tests also support that metastatic cells have an enhanced capacity to maintain a constant production of ATP under variable substrate availability.

Interestingly, the metabolic reprogramming we observed correlates with high levels of both MYC and E-cadherin in the metastatic cell lines compared to the primary tumour cell line. These results concur with the extensive existing literature on the metabolic changes driven by MYC in cancer cells [43,44] and with recent results demonstrating a role for E-cadherin as a promoter of metastasis and mitochondrial metabolism [45,46]. Recent findings on the differences between primary and metastatic colorectal cancer based on transcriptomic data indicate that metastasis is characterised by reduced EMT but increased MYC pathway activity [47]. Nevertheless, our metastatic cells also showed enhanced levels of mesenchymal markers (i.e., N-cadherin and vimentin) in comparison with the tumour primary cell line. Overall, our results suggests that the SW620 and LiM2 models underwent a mesenchymal-epithelial transition (MET) program from the primary tumour-derived cell line which had already lost E-cadherin and acquired some migratory capacity [48]. Indeed, it has been established that, in some cases, metastatic cells present epithelial features and that a transient EMT supports metastatic spread [49]. Additionally, the plasticity of EMT phenotype observed in the metastatic cell lines, especially in LiM2, could be related to the observed higher expression of SNAI1, a mesenchymal transcription factor associated with stemness [50,51].

GSMMs have long been used to identify putative metabolic targets against cancer [52,53,54]. Here, beyond the state of the art, we expanded the array of data incorporated into the model, which enhances reliability, by developing an approach integrating up to 7 layers of data (i.e., growth rates, rates of metabolite uptake and secretion, targeted metabolomics, ^13^C resolved metabolomics, transcriptomics and gene dependencies). This allowed the construction of highly accurate cell line specific GSMMs, which facilitated the identification of potential metabolic vulnerabilities associated with metastatic progression. Our cell line specific GSMMs predicted that the metastatic cell lines were dependent on cystine uptake from the extracellular media. Indeed, we determined that inhibition of the cystine transporter system xCT with sulfasalazine or erastin was highly specific for the metastatic cell lines, which could make it a potentially effective therapeutic strategy against metastasis in colorectal cancer. Indeed, the system xCT is overexpressed in several cancer types including colorectal tumours with mutant KRAS [55,56] and has been found to be strongly correlated with recurrence in colorectal cancer patients [57]. The system xCT plays a key role in supplying the cysteine, in the form of cystine, for glutathione synthesis [58] and, in this regard, GSR was also predicted to be a putative target since it is the enzyme that recovers reduced glutathione, which is necessary for the reduction of cystine to cysteine. The fact that GSR inhibition with 2-AAPA was selective for the metastatic cells validated this model prediction. In addition, we found a synergetic response when both de novo synthesis and recycling of glutathione were impaired by combining 2-AAPA and erastin treatments. Remarkably, in other cancer types, both system xCT and 2-AAPA inhibitors individually have already been proven to be successful in combination with cisplatin [59,60] or radiotherapy [61,62,63]. Thus, we posit that the combination of 2-AAPA and erastin could be used in conjunction with chemotherapy with electrophilic compounds (e.g., cisplatin) or radiotherapy to effectively tackle metastatic colon cancer.

Additionally, from genome-scale simulations, we also identified cytosolic folate metabolism as a pathway upregulated in the metastatic cell lines compared to the primary colorectal cell line SW480 and such an increase could not be solely attributed to increased cellular proliferation. Notably, while the inhibition of both the cytosolic and mitochondrial SHMT isoforms was effective for both primary and metastatic cancer cell lines, targeting only cytosolic activities by inhibition of DHFR or MTHFD1, inhibited proliferation selectively in the metastatic cell lines. This suggests that the metastatic cell lines could be selectively dependent on the cytosolic folate metabolism. Indeed, both DHFR and MTHFD1 have been found to be overexpressed in colorectal cancer tumours compared to healthy tumour epithelia [64]. Remarkably, folate metabolism is also intrinsically connected to cysteine and glutathione metabolism. For instance, it has been postulated that MTHFD1 supports metastatic spread in melanoma by acting as a source of NADPH, which can contribute to glutathione recycling and antioxidant capacity [65]. Indeed, the compartmentalisation predicted by our GSMMs could reflect the demand of NADPH in the cytosol, which cannot be compensated by the mitochondrial branch of folate metabolism. Additionally, folate metabolism also acts as one-carbon donor in the methionine cycle which contributes to the epigenetic modulation that is likely required to maintain the phenotype of metastatic cells [66]. Thus, we hypothesised that the combination of cystine uptake and folate metabolism inhibition could be effective against metastatic cells. Indeed, we found erastin and LY345899 to be a good combination that synergistically impaired cell proliferation as well as 3D growth in the metastatic cell lines.

Overall, our findings provide an insight into the metabolic reprogramming that supports the metastatic phenotype in a match patient primary tumour (SW480), lymph node (SW620) and liver-metastasis derived (LiM2) cellular model and unveil metabolic vulnerabilities that emerge during the transition from primary to metastatic states. Interestingly, there are only subtle differences between the metabolic phenotype of SW620 and LiM2 even though the latter are derived through in vivo selection of the most metastatic SW620 clones. This suggests that the cancer stem cell metabolic phenotype of lymph node metastatic cells allows them to adapt to any environment and metastasise. Indeed, this similarity can be therapeutically exploited as both populations were shown to share the same metabolic dependencies on cystine uptake and folate metabolism. Even more, we unveiled that the combination of existing drugs targeting both vulnerabilities is highly effective against such metastatic populations. 

While the SW480-SW620-LiM2 same-patient-derived model is uniquely suited to characterise the metabolic reprogramming underlying mCRC, colorectal cancer is a heterogeneous disease and the findings and vulnerabilities here reported might only apply to a subset of colorectal cancer tumours. Future works should explore the genetic histopathological and metabolic markers that can identify the CRC patients that could benefit the most by the putative drug or drug combinations here identified, thus paving the way for personalized therapies against mCRC [67]. Remarkably, methotrexate [37] and sulfasalazine [29] are already approved for clinical use and could be easily repositioned against susceptible subtypes of mCRC. Furthermore, the omics integration workflow here developed can easily be applied to the metabolic characterisation of other cancer cell models, potentially leading to the identification of targets against metastatic spread in other subtypes of mCRC or other cancer types.

## 4. Materials and Methods 

### 4.1. Cell Lines and Culture

SW480 cell line was obtained from the American Type Culture Collection (ATCC, Manassas, VA, USA). SW620 and its metastatic derivative SW620-LiM2 were obtained from Dr. Roger Gomis at IRB Barcelona [12], both lines were authenticated and KRAS mutation confirmed. NCM460 cell line [15] was a kind gift from Dr. Mary Pat Moyer (INCELL, San Antonio, TX, USA). All cells were grown in DMEM with 12.5 mM glucose, 4 mM glutamine, 5% Fetal Bovine Serum (10270, Gibco, ThermoFisher Scientific, Waltham, MA, USA) and 1% Streptomycin/penicillin at 37 °C in a 5% CO_2_ atmosphere. Cell volume was determined using a Scepter^TM^ Handheld Automated Cell Counter (Merk Millipore, Burlington, MA, USA).

### 4.2. Chemicals

LY345899, was purchased from Med Chem Express (Monmouth Junction, NJ, USA), SHIN2 was purchased from Glixx Laboratories (Hopkinton, MA, USA), methotrexate, sulfasalazine, 2-AAPA and erastin were purchased from Sigma-Aldrich (St. Louis, MO, USA).

### 4.3. Xenograft Experiments

SW480, SW620 and SW620-LiM2 cell lines (1 million cells/mouse) were injected subcutaneously into immunosuppressed NOD/SCID male mice, after a test for mycoplasma contamination. Before injections, the cells were trypsinised and resuspended in 50% Extracellular Matrix gel (E6909, Sigma-Aldrich, USA): 50% DMEM 12.5 mM Glc and 4 mM Gln, 5% FBS and 1% S/P. A total of 29 mice were injected, 10 mice per cell lines (except for LiM2, for which we injected 9 mice). Tumour volume was measured two times per week until the tumours reached 8–10 mm in diameter. All procedures were carried out at the Francis Crick Institute under pathogen-free conditions, in accordance with the Local Ethics Committee.

### 4.4. Cell Proliferation Assay Using Fluorospheres

Cell proliferation was determined by flow cytometry using flow-count fluorospheres (7547053, Beckman Coulter, Chaska, MN, USA). At the end of the incubation, cells were resuspended with media containing 100,000 fluorospheres/mL and immediately analysed by Gallios^TM^ Flow Cytometer (Beckman Coulter, Chaska, MN, USA). 

### 4.5. IC_50_ Curve Determination Using Hoechst

When determining IC_50_ using various concentrations of a specific drug or a combination on p96 well plates, cell proliferation was assessed by HO33342 staining; cells were washed with PBS, lysed with 0.01% SDS and frozen at −20 °C O/N, thawed at 37 °C and incubated with 4 μg/mL of HO33342 in 1 M NaCl, 1 mM EDTA, 10 mM Tris-HCl pH 7.4 for 1 h at 37 °C in the darkness. Fluorescence was measured at 460 nm after excitation with 337 nm in a FLUOstar OPTIMA Microplate Reader (BMG LABTECH GmbH, Ortenberg, Germany). 

### 4.6. Apoptosis Assay

Apoptotic cells were determined by flow cytometry using Annexin V coupled with fluorescent isothiocyanate (FICT). At the end of incubation, cells were resuspended in 10 mM HEPES pH 7.4, 140 mM NaCl, 2.5 mM CaCl_2_ buffer. Annexin V-FICT was added according to the kit’s instructions (Bender System MedSystem, Vienna, Austria) and an incubation was performed in the darkness during 30 min at room temperature. Propidium iodide was added at 20 μg/mL, 1 min before analysing at the GalliosTM Flow Cytometer (Beckman Coulter, USA). 

### 4.7. Spheroids Assays

Cell lines were seeded on 24-well (104 cells/well) low attachment plates in medium containing EGF, BFGF, heparin, B27, insulin and hydrocortisone and incubating for one week. Spheroids were analysed by phase-contrast microscopy and stained incubating them with 0.5 mg/mL MTT (3-[4,5-dimethylthiazol-2-yl]-2,5-diphenyltetrazolium bromide) for 4 h. Quantification was made by ImageJ software scanned images with the Analyse Particles tool, applying particle sizes from 0.0000785 to infinite cm^2^, and “total area” was taken as the value to estimate and compare spheroid formation capacity between samples. 

### 4.8. Wound Healing Assay

Cell lines were seeded on 24-well (6 × 10^5^ cells/well) in 24-plates. The media was replaced after 24 h by another media containing 0.5% of mitomycin and no Fetal Bovine Serum. After 1 h incubation, an artificial wound was performed by scratching the monolayer using a pipette tip. The wound’s width was measured at 0, 3, 7, 24 and 48 h using phase-contrast microscope images and ImageJ software. 

### 4.9. Western Blotting

Protein extracts were obtained from the cultured cells or dry tissue incubating for 20 min at 4 °C with RIPA buffer (50 mM Tris pH 8.0, 150 mM sodium chloride, 1% Triton X-100, 0.5% sodium deoxycholate, 0.1% sodium dodecyl sulphate, 1% protease inhibitor cocktail and 1% phosphatase inhibitor cocktail from Thermo Fisher Scientific Inc., Waltham, MA, USA) and scrapping, sonicating and centrifuging at 12,000× *g* for 15 min. Western blotting from equal amounts of protein extracts was performed as explained in Appendix A. The primary antibodies used are specified in Appendix A. Quantification of the bands was performed using ImageJ software.

### 4.10. Immunohistochemistry

Tumour tissue, liver, lung and spleen were obtained from mice wearing a tumour of 8–10 mm in diameter (see Xenograft experiments section). Deparaffinated tissues in slides where processed for immunohistochemistry according to the Dako kit EnVision K4065 instructions, specified in Appendix A. Images of the stained slides were taken in an optical microscope (40×) and analysed with ImageJ software.

### 4.11. Spectrophotometric Measurements

Glucose, lactate, glutamine and glutamate concentrations in the cell culture media at initial and final incubation time were measured using a COBAS Mira Plus spectrophotometer (Horiba ABX, Kyoto, Japan). Such determinations are specified in Appendix A. 

### 4.12. OCR Measurements, Mito Stress and Mito Fuel Assays

Oxygen consumption rates were measured using a Seahorse XF24 Flux Analyzer (Seahorse Bioscience, USA). Cells were seeded at 7.5 × 10^4^ cells/well density for SW480 and 10^5^ cell/well density for SW620 and LiM2 in 24-well plate pre-coated with collagen (Advanced Biomatrix). The assays were conducted according to the manufacturer’s instructions, see Appendix A. 

### 4.13. Targeted Metabolomics 

Intracellular and extracellular metabolite profiling was performed using the Absolute IDQTM p180 kit (Biocrates Life Sciences AG, Tyrol, Austria) according to the manufacturer’s instructions, see Appendix A. 

### 4.14. Stable Isotope-Resolved Metabolomics In Vitro

Quantities of 2.5 × 10^6^, SW480, SW620 and LiM2 cells were seeded in 100 mm plates and after 24 h the media was changed to either glucose 50% enriched in [1,2-^13^C]-glucose, glutamine 50% enriched in [U-^13^C]-glutamine, or unlabelled substrates. Cells and media were obtained for metabolite extractions at 6 and 24 h after the labelled substrates were added. The metabolites analysed and the procedures are specified in Appendix A.

### 4.15. Stable Isotope-Resolved Metabolomics In Vivo

After cell line injections into immunocompromised NOD/SCID mice (see Xenograft experiments section) and once tumours reached 8–10 mm in diameter mice were given a bolus of either [U-^13^C]-glucose (1 bolus of 20 mg/35 g) or [U-^13^C]-glutamine (2 boluses of 6 mg/35 g each, with 15 min interval). Fifteen minutes after the last bolus, the mice were terminally anaesthetised, and blood was taken by cardiac puncture. Tumour tissue was snap frozen in liquid nitrogen. Then, tumour samples were ground in liquid nitrogen to form a powder, which was lyophilised. Polar intracellular metabolites were analysed as specified in Appendix A. 

### 4.16. Statistical Analyses

All experiments were performed at least in triplicates and repeated two or more times. Statistical analyses of experimental measures were performed using the Agricolae package for R. More specifically, for comparisons that are between the cell lines (SW480–SW620–LiM2) we used a one-way ANOVA for the factor “cell line”, and Scheffe’s test for multiple comparisons. Groups that are indicated with the same letter are not significantly different (*p* > 0.05). For comparisons between two conditions (e.g., before and after drug administration), we used Student’s *t* test with *p* < 0.05, and we used asterisk to indicate significant differences. Further statistical analyses are explained in Appendix A.

### 4.17. Multiomics Data Integration

Multiple layers of data were integrated to compute cell line specific flux maps using a novel workflow (Figure 4). The workflow involved the following series of steps, first a central carbon metabolism flux map was computed using ^13^C MFA [68], and then integrated into the human GSMM model Recon2 [69]. Next, targeted metabolomics were integrated in order to constrain GSMMs to produce intracellular metabolites at a rate proportional to their concentrations and the proliferation rate of each cell line [70]. Then, Minimal Cut Set Analysis (MCS) was used to integrate reported essential metabolic genes from project DRIVE [20] (Appendix A). Afterwards, within the above-defined constraints, the GIMME algorithm [71,72] was used to restrict the maximum flux through reactions based on transcript expression evidence. Finally, flux sampling was used to compute flux combinations consistent with the above-integrated data. For each of the flux samples, gene KOs were simulated, and flux samples were ranked based on their consistency with gene dependency data. The latter step served both to (i) minimise the false positives of gene essentiality, and (ii) integrate partial dependencies on non-essential genes. The average of the top 100 ranked flux samples were selected as the cell line-specific flux maps. A detailed description of each step is provided in Appendix A.

### 4.18. Identifying Putative Metabolic Targets

To identify metabolic targets against colon cancer, gene KOs were systematically simulated for metabolic genes, individually or in pairs. The reactions to be blocked by each gene KO(s) were determined by combining gene expression data with the GPR rules of Recon 2. A reaction was considered to be inactive if when a gene was inactivated (i.e., its expression set to 0), the mapped gene expression value decreased at least 16-fold. Then, the effect of reaction KOs was simulated using the reference flux distribution computed for each cell line as input for running MOMA [21] in the framework of cell line specific models. Single gene KOs were systematically performed for all cell lines under study. Conversely, for gene pairs, due to the larger number of combinations to test (~175,000), all potential gene pairs were only evaluated in SW620. Next, the gene combinations that resulted in a biomass production below 15% of wild type in SW620 and displayed synergy were evaluated on the remaining cell lines. A gene pair was considered to have synergy if the fraction of biomass production under the double KO was less than the product of the fraction of biomass production under the single KOs. A gene or gene pair was considered a potential target if it reduced the biomass production to 10% or less of the wild type in both SW620 and LiM2. Some of the identified targets were potentially inherent vulnerabilities to all human metabolic networks (as opposed to only cancer specific metabolic networks). With this in mind, all single and gene pair combinations were simulated on a Recon2 network unconstrained by condition specific data (e.g., transcriptomics, metabolomics and gene dependency) and the targets that prevented synthesis of biomass components in this generic network were filtered out.

## 5. Conclusions

Using a novel systems biology strategy to integrate metabolic and transcriptomic data, we unveiled the metabolic reprogramming related to cystine transport and folate metabolism as druggable metabolic vulnerabilities of mCRC in the SW480-SW620-LiM2 same-patient-derived model. We demonstrated that the combined inhibition of xCT and MTHFD1 is synergetic and is specific for metastatic colon cancer cell lines.

## Figures and Tables

**Figure 1 cancers-13-00425-f001:**
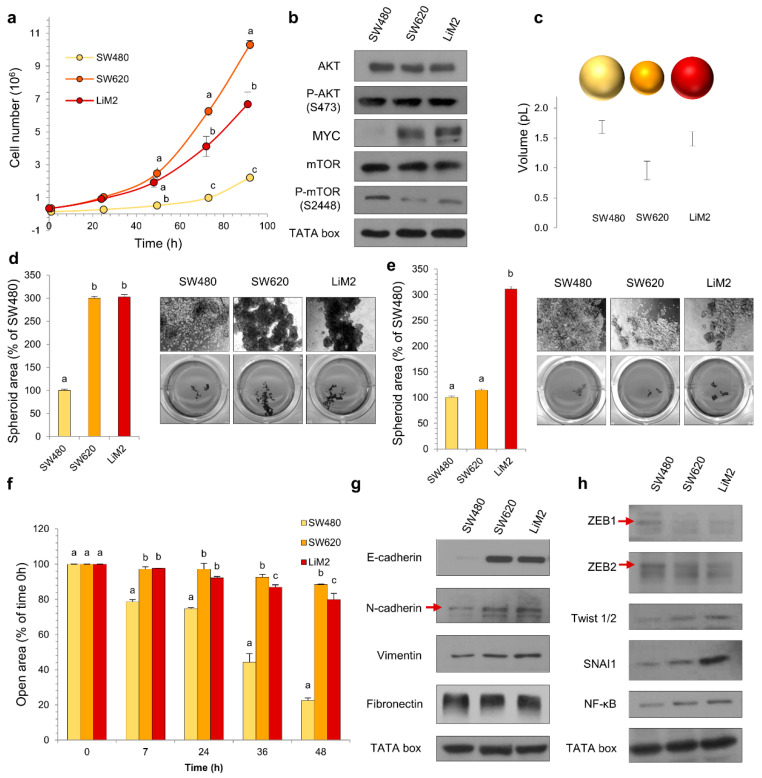
Characterisation of the metastatic phenotype. (**a**) Growth curve from 0 to 96 h incubation with DMEM 12.5 mM Glc and 4 mM Gln, 5% FBS and 1% S/P. (**b**) Protein levels of the main oncogenic signalling pathways tested by western blotting. TATA box was used as a loading control. (**c**) Cell volume measured by Scepter^®^. (**d**) Spheroid formation assay and (**e**) secondary spheroid formation assay. Quantification of spheroid area in the left. Images of contrast-phase microscope (40×) in the right and scan (1×). (**f**) Migration area quantification from a wound healing assay. (**g**) and (**h**) Protein levels of EMT markers and related transcription factors tested by western blotting. ^a,b,c^ A one-way ANOVA and Scheffe’s test for multiple comparisons was performed for the factor “cell line”.

**Figure 2 cancers-13-00425-f002:**
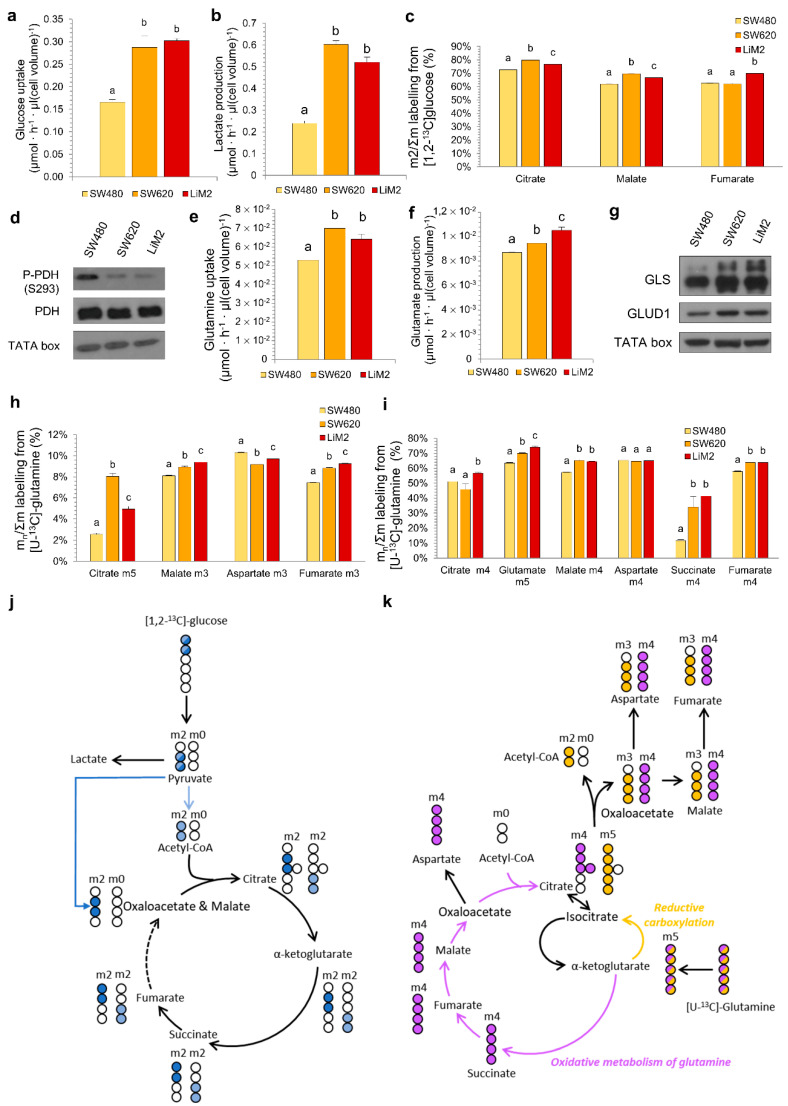
The metastatic cell lines display higher metabolic flexibility with an increased glucose and glutamine metabolism. (**a**) Glucose uptake and (**b**) lactate production rates after 48 h of cell culture. (**c**) Glucose contribution to TCA cycle after 24 h incubation with 10 mM [1,2-^13^C]-glucose. (**d**) P-PDH (phosphorylated Pyruvate Dehydrogenase) and total PDH protein levels tested by western blotting. TATA box was used as a loading control. (**e**) Glutamine uptake and (**f**) glutamate production rates after 48 h of cell culture. (**g**) GLS (glutaminase), GLUD1 (glutamate dehydrogenase 1) protein levels tested by western blotting. (**h**) and (**i**) Glutamine contribution to TCA cycle measured after 24 h incubation with 4 mM [U-^13^C]-glutamine, representing (**h**) reductive carboxylation and (**i**) oxidative metabolism of glutamine. (**j**) Graphical representation of the ^13^C (in blue) and ^12^C (in white) incorporation to TCA intermediaries from [1,2-^13^C]-glucose. Only the formation of m2 isotopologues in the first turn of the cycle is shown. (**k**) Graphical representation of the ^13^C (coloured) and ^12^C (in white) distribution to intracellular TCA intermediaries by oxidative carboxylation (purple) or reductive carboxylation (orange) from [U-^13^C]-glutamine in the first turn of the cycle. ^a,b,c^ A one-way ANOVA and Scheffe’s test for multiple comparisons was performed for the factor “cell line”.

**Figure 3 cancers-13-00425-f003:**
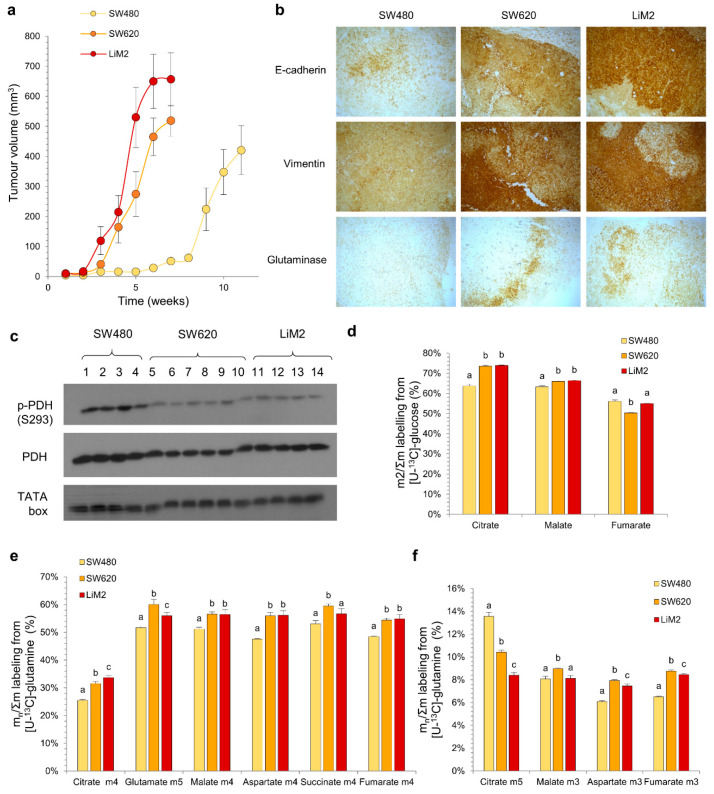
The metabolic adaptation of the metastatic cell lines observed in vitro is maintained in an in vivo scenario. (**a**) Tumour volume evolution measured by calliper. (**b**) Immunohistochemical staining of paraffin-embedded tumour slides (40×). (**c**) P-PDH (phosphorylated pyruvate dehydrogenase at S293) and PDH (total pyruvate dehydrogenase) protein levels tested by western blotting of tumour extracts. TATA box was used as a loading control. Samples 1–14 indicate extracts from different mice that were injected with SW480 (1–4), SW620 (5–10) or LiM2 (11–14). (**d**) Glucose contribution to TCA cycle measured from a tumour extract after a 15-min bolus of [U-^13^C]-glucose (20 mg/35 g) before the mice were culled. (**e**) and (**f**) Glutamine contribution to TCA cycle measured from a tumour extract after two boluses of [U-^13^C]-glutamine (6 mg/35 g, with 15 min interval) before the mice were culled, representing (**e**) oxidative metabolism of glutamine and (**f**) reductive carboxylation. ^a,b,c^ A one-way ANOVA and Scheffe’s test for multiple comparisons was performed for the factor “cell line”.

**Figure 4 cancers-13-00425-f004:**
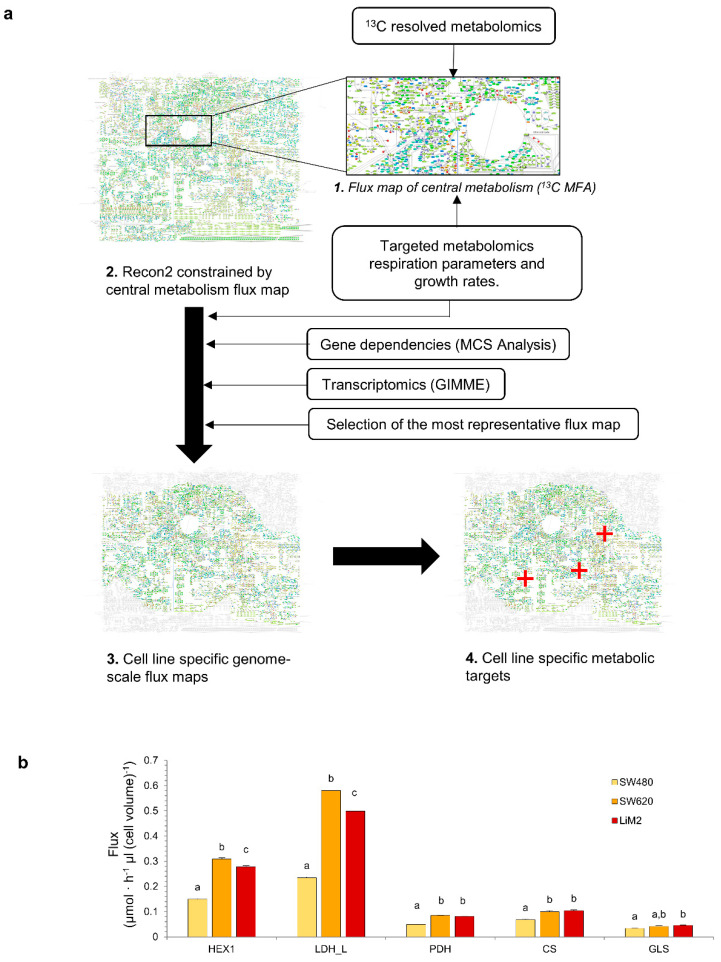
Computational inference of cell line-specific metabolic flux maps and metabolic targets through multiomics data integration. (**a**) Graphical representation of the multiomics integration workflow. First, a central carbon metabolism flux map is estimated using ^13^C MFA, next the flux map is used to constrain the generic human GSMM Recon2, then several layers of omics data are used to estimate the cell line specific flux maps which can be used to identify metabolic vulnerabilities to selectively target specific populations. (**b**) Predicted Fluxes through HEX1 (hexokinase), LDH-L (lactate dehydrogenase), PDH (pyruvate dehydrogenase), CS (citrate synthase) and GLS (glutaminase). ^a–c^ denote cell lines with an overlap of the sampled flux values for a given reaction.

**Figure 5 cancers-13-00425-f005:**
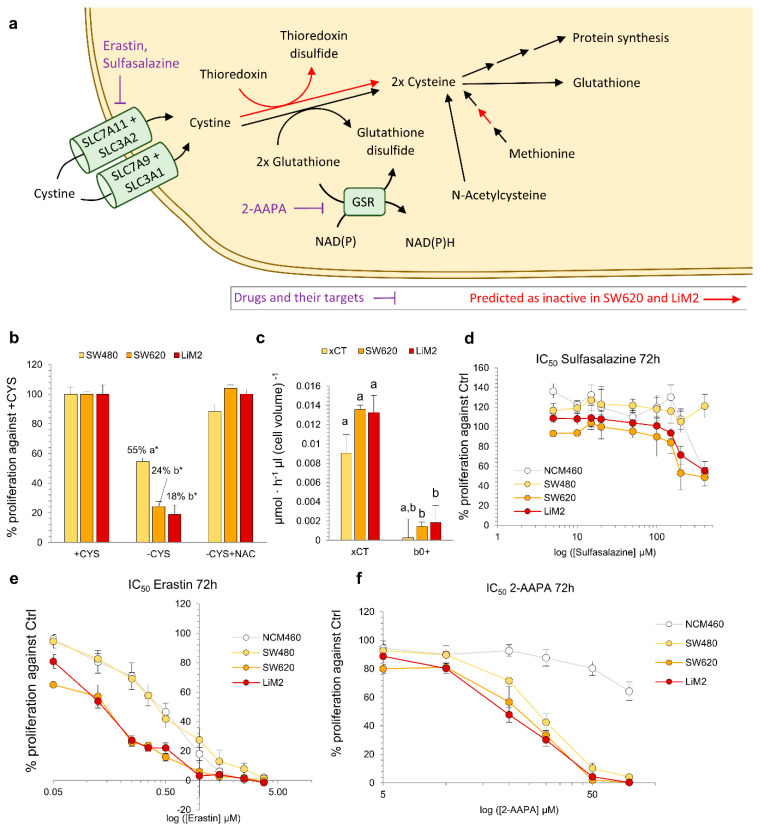
Metastatic cells are dependent on cystine uptake and vulnerable to system xCT and glutathione reductase inhibition. (**a**) Representation of cystine and glutathione metabolism. (**b**) Cell proliferation measured by DNA content using HO33342 under cystine deprivation (-CYS) and adding N-acetylcysteine (-CYS+NAC). *Student’s *t* test for –CYS or –CYS+NAC vs. Control conditions, *p* < 0.05. ^a,b^ A one-way ANOVA and Scheffe’s test for multiple comparisons for the factor “cell line”. (**c**) Predicted fluxes through the system xCT and b^0,+^ system, ^a–c^ denote cell lines and reactions with an overlap of the sampled flux values for a given reaction. (**d**) and (**e**) Cell viability curve for (**d**) sulfasalazine (system xCT inhibitor), (**e**) erastin (system xCT inhibitor) and (**f**) 2-AAPA (GSR inhibitor) assessed by DNA content after 72 h incubation. Statistical analyses of the IC_50_ curves are shown in Appendix A.

**Figure 6 cancers-13-00425-f006:**
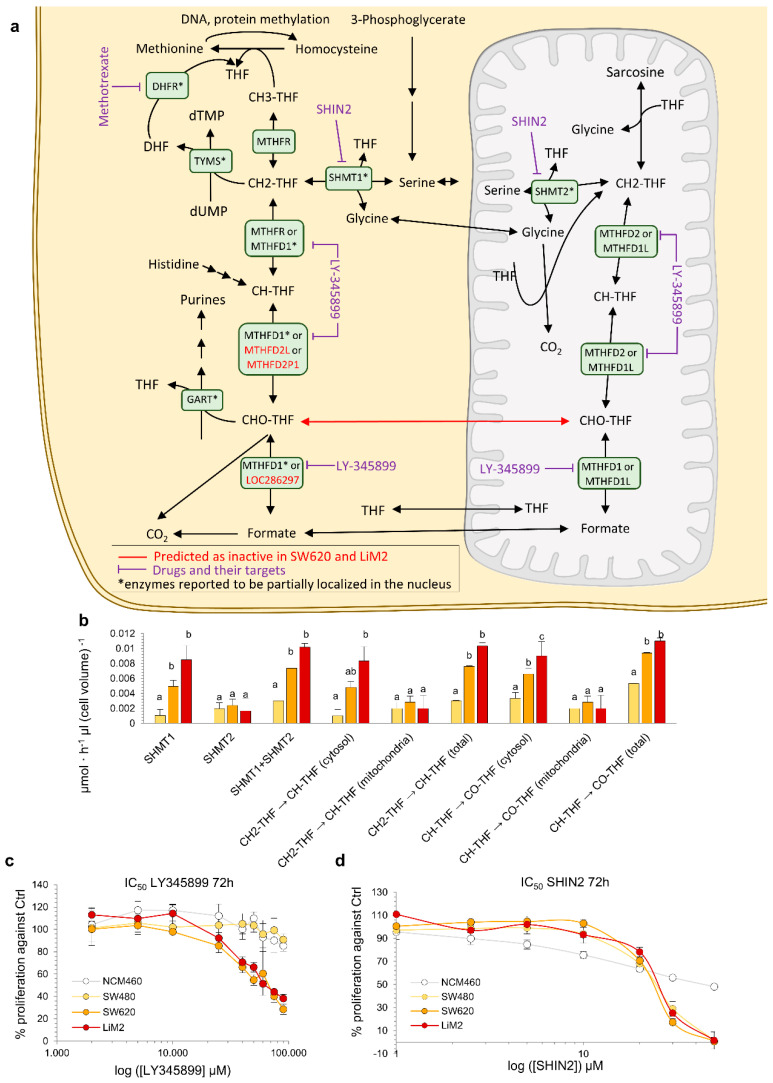
The metastatic cell lines are dependent on the cytosolic branch of folate metabolism. (**a**) Representation of folate metabolism. (**b**) Predicted flux values for different reactions of cytosolic and mitochondrial folate metabolism. ^a,b,c^ denote cell lines with an overlap of the sampled flux values for a given reaction. (**c**,**d**) Cell viability curve for the MTHFD1/2 inhibitor LY345899 (**c**) or the SHMT1/2 inhibitor SHIN2 (**d**) determined by DNA content using HO33342 after 72 h incubation. Statistical analyses of the IC_50_ curves are shown in Appendix A.

**Figure 7 cancers-13-00425-f007:**
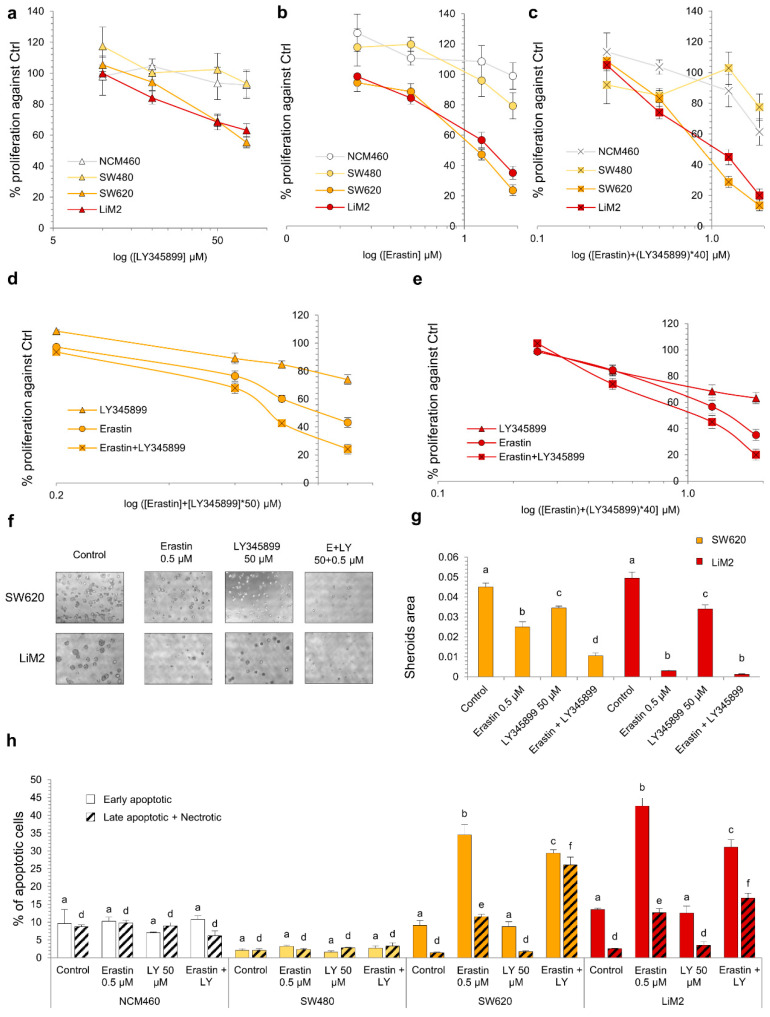
Synergistic effect of the simultaneous inhibition of cystine uptake and folate metabolism. (**a**–**c**) Cell viability curve for LY345899 (**a**), erastin (**b**) and the combination of both inhibitors (**c**) assessed by DNA content after 72 h incubation. (**d**,**e**) Same experiment in (**a**–**c**) comparing the three treatments (erastin or LY345899 individually and the combination) for the different cell lines. (**f**,**g**) Spheroid formation assay with erastin (0.5 μM), LY345899 (50 μM) or the combination. (**f**) Images of contrast-phase microscope (40×) and (**g**) quantification of spheroid area from scanner images. (**h**) Percentage of early apoptotic cells measured by flow cytometry using Annexin V-PI under erastin (0.5 μM), LY345899 (50 μM) or the combination after 72 h incubation. ^a–f^ A one-way ANOVA and Scheffe’s test for multiple comparisons was performed for the factor “drug treatment”. The statistical analyses for the drug combinations are shown in Appendix A.

**Table 1 cancers-13-00425-t001:** Most promising metabolic targets identified from the cell-specific GSMMs. List of the individual paired target genes and their simulated effect on growth rate, expressed as percentage of growth of the simulated KO vs. wild type of the same cell line.

Gene KO(s)		Predicted Fraction of Growth Compared to Wild Type		
		SW480	SW620	LiM2
Single Targets	*MTHFD1*	100%	0%	0%
	*GSR*	99%	0%	0%
Target Pairs	*SLC7A9*, *SLC3A2*	86%	0%	0%
	*SLC3A1*, *SLC3A2*	86%	0%	0%
	*SLC7A9*, *SLC7A11*	85%	0%	0%
	*SLC7A11*, *SLC3A1*	85%	0%	0%

## Data Availability

The datasets created and analysed in the present study are available from the corresponding author on reasonable request.

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
