# Peer review of "Cysteine and Folate Metabolism Are Targetable Vulnerabilities of Metastatic Colorectal Cancer"

_cancers, 2021, doi:10.3390/cancers13030425_

Round 1

Reviewer 1 Report

In the manuscript entitled “Cysteine and folate metabolism are targetable vulnerabilities of metastatic colorectal cancer”, Tarragó-Celad and co-authors report an interesting study in which they identified metastatic cell-specific metabolic vulnerabilities in colorectal cancer. Integrating multi-omics technologies, they identified cysteine and folate metabolic pathways as potential therapeutic targets in metastatic colorectal cancer.

From the chemotherapeutic standpoint, the article is not very novel because different strategies have been proposed in the last few years to target both the one-carbon metabolism and cysteine/cystine uptake to reduce nucleotide and glutathione biosynthesis. However, the proposed drug combinations could be still relevant to reduce colon cancer mortality most of the proposed drugs already approved for other diseases.

The investigators report a massive amount of data in the text and in the SI that are not always easy to visualize and understand. The manuscript is not well organized and the reader struggles to follow the experimental design.

For readers not familiar with metabolic flux analysis, I strongly encourage including metabolic pathways (e.g. figures 1 and 2) showing clearly how the level of 13C isotopomers from labeled glucose and glutamine are distributed following chemotherapeutic treatment.

Metabolic data do not strongly support the metabolic vulnerabilities of the folate metabolic pathway. Except for the choice of methotrexate, there are no clear metabolomics data suggesting this pathway is downregulated following treatment and how different cell lines are more susceptible to the proposed drug combination (either synergistically and additively).

Instead of depriving glucose or glutamine, probably a therapeutic intervention would be more relevant from a chemotherapeutic strategy. Also, experiments injecting alternative substrates (arginine, leucine, proline, glycine, serine, and uridine) are not clearly explained and followed up with additional experiments (e.g. using different isotopes). Not sure, if they help the manuscript.

I do not believe normal cell lines are used as control; this might be relevant to understand the selectivity of the proposed drugs.

Also for the in vivo experiments, I struggle to find the level of cysteine and cystine in both blood and tissue samples. This could help to understand if the depletion of these metabolites has been reached in the in vivo experiments.

Reviewer 2 Report

Tarrago-Celada et al. compared three colorectal cancer cell lines (one non-metastatic and two metastatic) for their growth and metabolic properties. A computational work flow analysis was performed incorporating the metabolomics and transcriptome data, and identified glutathione homeostasis and folate pathway as vulnerable targets for treating the metastatic cancers. The topic of identifying metabolic and transcriptional difference and vulnerability between non-metastatic and metastatic cancers is important and interesting. The work contains a large amount of data and the integrated omics analysis is quite interesting. There are several major drawbacks: 1) the three cell lines are claimed to be metastatic vs non-metastatic, yet there is no metastatic properties associating with these cell lines. 2) there is a cause/effect issue regarding the different growth and metabolic properties. All metabolic differences may just be a consequence of different growth rates of the cell lines. 3) The three cell lines are not truly syngeneic hence difficult to dissect what may be contributing to the different growth and metabolic properties. 4) Comparing just three cell lines (one non-metastatic) is not enough to generate general conclusions. 5) There is no statistical analysis of all figures. 6) The manuscript is often written in a tedious way and very difficult to follow. If the authors can reasonably address these issues this manuscript may be worth publishing with its value in attempting to establish a metabolic connection in metastatic cancer growth and treatment.

Reviewer 3 Report

The manuscript focuses on primary colon adenocarcinoma, lymph node, and metastatic liver cancer from the same patients. The data are presented very nicely regarding cysteine and folate metabolism and specifically metastatic cells sensitivity about cystine import and folate metabolism.  

Minor editorial/spell corrections are required. For example, in Figure 7, the spheroid is misspelled.

Round 2

Reviewer 1 Report

The authors have significantly improved the previously submitted manuscript and I agree with most of their edits.

Reviewer 2 Report

The authors have adequately addressed previous concerns.